# Polydopamine-Modified Al_2_O_3_/Polyurethane Composites with Largely Improved Thermal and Mechanical Properties

**DOI:** 10.3390/ma13071772

**Published:** 2020-04-09

**Authors:** Ruikui Du, Li He, Peng Li, Guizhe Zhao

**Affiliations:** 1North University of China, Taiyuan 030051, China; zminggu@163.com (L.H.); lpzbdx@163.com (P.L.); 2Shanxi Province Polymer Composite Engineering Technology Research Center, Taiyuan 030051, China

**Keywords:** polyurethane, alumina, thermal conductivity, mechanical properties, in situ polymerization

## Abstract

Alumina/polyurethane composites were prepared via in situ polymerization and used as thermal interface materials (TIMs). The surface of alumina particles was modified using polydopamine (PDA) and then evaluated via Fourier transform infrared spectroscopy (FTIR), thermogravimetric analysis (TG), and Raman spectroscopy (Raman). Scanning electron microscope (SEM) images showed that PDA-Al_2_O_3_ has better dispersion in a polyurethane (PU) matrix than Al_2_O_3_. Compared with pure PU, the 30 wt% PDA-Al_2_O_3_/PU had 95% more Young’s modulus, 128% more tensile strength, and 76% more elongation at break than the pure PU. Dynamic mechanical analysis (DMA) results showed that the storage modulus of the 30 wt% PDA-Al_2_O_3_/PU composite improved, and the glass transition temperature (Tg) shifted to higher temperatures. The thermal conductivity of the 30 wt% PDA-Al_2_O_3_/PU composite increased by 138%. Therefore, the results showed that the prepared PDA-coated alumina can simultaneously improve both the mechanical properties and thermal conductivity of PU.

## 1. Introduction

With the improvement of the miniaturization of electronic products, electronic packaging has led to higher requirements for thermal management of thermal interface materials. Therefore, efficient heat dissipation has become a key issue to improve the high-performance service life of electronic devices. Many researchers have now used polymer substrates as thermal interface materials [1,2,3]. Polyurethane (PU) has excellent adhesion, low-temperature resistance, high elasticity, vibration damping and toughness, etc. [4,5]. Its molecular chain is composed of a soft segment and hard segment. PU can be prepared by adjusting the ratio of the hard segment and soft segment due to its excellent comprehensive performance. The soft segment endows the material with better characteristics of being easy to stretch and retract, while the hard segment endows the material with good resilience and strength [4,6]. In order to improve the low intrinsic thermal conductivity of polyurethane (~0.20 W/m K), certain thermal conductivity filler is usually considered to put into polyurethane to improve thermal conductivity [5,7].

Common thermally used conductive fillers are metals (Ag [8] and Cu [9]), carbon-based (graphene [10], carbon nanotubes [11]), and ceramics (BN [12], Al_2_O_3_ [13], and AlN [14,15]). Metal fillers and carbon-based fillers also increase electrical conductivity while improving mechanical and thermal properties; however, they are disadvantageous for Insulation properties of thermal interface materials. Compared with the other two fillers, ceramic fillers have the least influence on conductivity [16,17,18]. In the field of electronic packaging, inexpensive, easy-to-process ceramic fillers are excellent candidates with high thermal conductivity and excellent electrical insulating properties. Compared with other ceramic fillers, spherical Al_2_O_3_ has often been used as electronic packaging materials because it exhibits higher thermal conductivity (32 W/m K), easier surface modification, wider source, higher load, lower viscosity, better fluidity, and excellent insulation performance [19,20]. Alumina as a thermal conductivity filler to enhance the thermal conductivity of polymers has been widely examined by many researchers [21,22,23].

In general, the high thermal conductivity of polymer composites mainly depends on whether the filler has a good thermal conduction path or network, which is mainly related to the interfacial interaction and dispersion of the filler in polymers matrix [7,24,25,26]. Generally speaking, poor interface combination between a thermal conductivity filler and polymer matrix leads to poor dispersion of the filler. These all cause strong phonon scattering and reduce phonon transmission efficiency, and then lead to the poor thermal conductivity of the interface [12,18,20,27,28]. It is reported that surface modification and functionalization of the filler can improve compatibility with the polymer matrix [29,30]. Common surface modification methods are hydroxylation [31], surfactant [32], coupling agent [27,33], etc. In recent years, bio-activated modification of polydopamine has become popular as a simple, environmentally-friendly, and efficient method. Polydopamine (PDA) is easy to adhere to the surface of inorganic and organic materials, and the abundant phenolic hydroxyl groups on the surface of polydopamine are conducive to the formation of stronger interaction with the resin matrix [9,13,17,34]. In this study, the surface of spherical Al_2_O_3_ was modified using polydopamine as a modifier. In order to realize insulating composites with high thermal conductivity and mechanical properties, a high load of spherical Al_2_O_3_ particles are required. However, if the filler cannot be uniformly dispersed in the matrix and has good compatibility with the matrix, accumulation and pores will be formed, which will reduce the mechanical properties of the composites [12,33,35]. Therefore, surface modification of the thermal conductivity filler under a high load can not only improve the thermal conductivity of the composites but also improve the mechanical properties. WonduDet et al. [16] used γ-Aminopropyltriethoxysilane (APTES) to modify Al_2_O_3_ and then added it to a thermoplastic polyurethane matrix (TPU) to prepare a thermally conductive composite by melt extrusion. The results show that compared with pure polyurethane, the thermal conductivity, Young’s modulus, and storage modulus of the composites increased by 80.6%, 55%, and 132%, respectively. However, under a high load, its mechanical properties have not been improved. Therefore, it is necessary to prepare composites with high thermal conductivity and high mechanical properties even under a high load.

In this paper, Al_2_O_3_/PU and PDA-Al_2_O_3_/PU composites were prepared via in situ polymerization. The thermal properties, mechanical properties, and morphology of the composites were characterized and evaluated via a thermal conductivity meter (TC), dynamic mechanical analysis (DMA), universal testing machine, and scanning electron microscope (SEM).

## 2. Materials and Methods

### 2.1. Materials

Spherical α-Al_2_O_3_ particles (average diameter of 3 μm) were purchased from Qinhuangdao Yinuo High-tech Materials Development Co., Ltd (Qinhuangdao, China). A two-part polyurethane casting resin was provided by Shanxi Coal Chemical (Taiyuan, China). Part A was a polyester-type hydroxy compound. Part B was the adduct of polyisocyanate (TDI) and glycerin, whose end group was a prepolymer of -CNO. Tris-(hydroxymethyl)-amino-methane (Tris, 99.9%) and Polydopamine hydrochloride (PDA, 98%) were purchased from Aladdin. SEM images of pristine Al_2_O_3_ are in Figure 1.

### 2.2. Modification of Al_2_O_3_ Particles with PDA

The scheme of modification on an Al_2_O_3_ surface with polydopamine is shown in Figure 2. The Al_2_O_3_ particles were dried at 80 °C for 12 h. First, 0.6 g of tris was added to 100 mL of distilled water and HCl to form a Tris-HCl buffer, and the pH was adjusted to 8.5. Then, 10 g of the dried Al_2_O_3_ particles was added to the above buffer solution, and the Al_2_O_3_ particles were uniformly dispersed by ultrasonic probe for 15 min., and then 1 g of PDA was added. The suspension was stirred magnetically at 25 °C for 24 h. It was found that the color of the suspension changed from brown to black gradually. Finally, the modified alumina particles (PDA-Al_2_O_3_) were collected by suction filtration, then washed three times with distilled water and dried at 80 °C for 24 h.

### 2.3. Preparation of Al_2_O_3_/PU and PDA-Al_2_O_3_/PU Composites

Polyurethane composites were prepared via in situ polymerization as illustrated in Figure 3. Polyurethane part A was weighed in a beaker, and the required filler was weighed on a weighing paper. Then, Al_2_O_3_ or PDA-Al_2_O_3_ particles were added to Part A via ultrasonic stirring for 90 min to evenly disperse them. Then, Part B was added to the above system and stirred continuously (the mass ratio of Part A and Part B was 1:1 as specified by the manufacturer). Finally, the mixed material was vacuum defrothed and cast at 80 °C for 8 h to obtain composites. The formed composites were pure polyurethane, 10 wt% pristine alumina/polyurethane, 10 wt% polydopamine modified alumina/polyurethane, 20 wt% pristine alumina/polyurethane, 20 wt% polydopamine modified alumina/polyurethane, 30 wt% pristine alumina/polyurethane, 30 wt% polydopamine modified alumina/polyurethane, which we abbreviate as PU, 10 wt% Al_2_O_3_/PU, 10 wt% PDA-Al_2_O_3_/PU, 20 wt% Al_2_O_3_/PU, 20 wt% PDA-Al_2_O_3_/PU, 30 wt% Al_2_O_3_/PU, and 30 wt% PDA-Al_2_O_3_/PU, respectively. All percentages are weight percentages (wt/wt%).

### 2.4. Methods

The surface functional groups of Al_2_O_3_ and PDA-Al_2_O_3_ particles were analyzed by Fourier transform infrared spectroscopy (FTIR; Nicolet, IS50, Thermo Fisher Scientific, Waltham, MA, USA). The sweep frequency range was 4000–400 cm^−1^. FTIR uses the KBr tablet pressing method. The resolution is 4 cm^−1^, and the number of scans is 16. The surface functional groups of Al_2_O_3_ and PDA-Al_2_O_3_ particles were analyzed using a micro Raman spectrometer (inVia Reflex; Renishaw, London, England). The laser wavelength is 532 nm in Raman.

The thermal conductivity of the composites was analyzed using a flat plate thermal conductivity meter (DRPL-II, Xiangtan Xiangyi Instrument Co., LTD, Xiangtan, China). The temperature of the hot plate and cold plate of the thermal conductivity meter is 30 °C and 50 °C, respectively. The sample size is a wafer with a diameter of 60 mm and a thickness of about 2 mm. The number of samples tested for thermal conductivity was 3.

The thermal stability of PDA-Al_2_O_3_ or composites was analyzed via thermogravimetric analysis (TGA; Q50, TA Instruments, Newcastle, Delaware, USA). About 6 mg of each sample was taken and heated to a temperature of 800 °C at a heating rate of 10 °C/min. The TGA experiments were carried out under a nitrogen atmosphere.

The mechanical properties of the composites, namely, tensile strength, Young’s modulus, and elongation at break, were tested using a universal testing machine (AI-7000s; Taiwan High Speed Rail Testing Instrument Co., Ltd., Taiwan, China). According to GB/T 528-2009, the loading speed was 100 mm/min at room temperature (https://www.doc88.com/p-6641254359663.htm). Samples had dumbbell shapes with about 6 mm width and 100 mm gauge length, and thicknesses were about 2 mm. The number of samples tested for mechanical properties was 5.

The storage modulus and loss angle tangent (tanδ) of the composites were characterized using dynamic mechanical analysis (DMA, TA Q800, Newcastle, DE, USA) in double cantilever mode at a frequency and temperature range of 1 Hz and −70 to 150 °C, respectively. The length, width, and height of the sample are 10, 5, and 5 mm, respectively, in DMA.

The dispersion of the particles in the composites was analyzed after fracturing all the specimens by liquid nitrogen, using field emission scanning electron microscopy (SEM; SU8000, Hitachi, Tokyo, Japan) after coating all specimen layers with gold to inhibit accumulation of charges. The surface element composition was analyzed by an X-ray energy spectrum (EDS) equipped with SEM.

The water absorption of Al_2_O_3_/PU and PDA-Al_2_O_3_/PU composites was calculated. According to GB/T 1462-2005, the dry sample was immersed in water at 23 °C for 24 h to test the water absorption percentage relative to the sample mass [36]. The average of the water absorption of five samples was the final water absorption. The formula for calculating the water absorption is shown below.
(1)$$\mathrm{Water}\;\mathrm{absorption}=\frac{M_{1}-M_{0}}{M_{0}}\times 100\%$$
where M_1_ is the quality of the sample after absorbing water (g), and M_0_ is the sample’s initial quality (g).

Descriptive statistics all use average plus or minus standard deviation.

## 3. Results and Discussion

### 3.1. Surface Modification

The FTIR spectra for the Al_2_O_3_ and PDA-Al_2_O_3_ are shown in Figure 4.

Compared with pristine Al_2_O_3_, some new peaks appear in the spectrogram of PDA-Al_2_O_3_. Due to the presence of O-H and N-H on the PDA attached to the alumina surface, there is a wide stretching vibration peak of PDA-Al_2_O_3_ around 3280 cm^−1^. The peak at 1602, 1516, and 1289 cm^−1^ corresponds to the stretching vibration of C = C in benzene ring, the bending vibration of N-H, and stretching vibration of aliphatic amine C-N, respectively [13]. The results show that PDA has been modified on Al_2_O_3_ successfully.

The thermal stability of pristine Al_2_O_3_ and PDA-Al_2_O_3_ particles is shown in Figure 5.

PDA-Al_2_O_3_ begins to decompose at about 220 °C because the PDA coating on the surface of Al_2_O_3_ begins to bedecomposed [37]. By comparing the difference in mass loss at 800 °C, PDA coated on the surface of Al_2_O_3_ is about 7.01%. Due to its excellent chemical stability, the mass loss of PDA-Al_2_O_3_ is small in the range from room temperature to 800 °C.

Figure 6 shows the significant difference between pristine Al_2_O_3_ and PDA-Al_2_O_3_ in the Raman spectra.

Due to the vibration mode of the Al-O, a sharp characteristic peak appears at 472 cm^−1^. Due to tensile vibration and deformation of catechol in the polydopamine poly structure, PDA-Al_2_O_3_ exhibits two broad peaks at 1355 and 1569 cm^−1^ [37,38].

### 3.2. Morphology

In order to directly observe the dispersion of the particles in the composites, SEM photos were taken, and the EDS of aluminum in the composites was tested.

In Figure 7a,c, Al_2_O_3_ particles are accumulated and exposed on the fracture surface, which indicates the poor interface adhesion between Al_2_O_3_ particles and the PU matrix, and the poor dispersion performance. In Figure 7b,d, PDA-Al_2_O_3_ has a better dispersion and smoother fracture surface in PU. This is because the compatibility between the particles and the matrix is improved by the modification with polydopamine. SEM images of composites with other contents are presented in Appendix A. We can find that alumina-filled composites with different contents can also lead to similar conclusions to those above.

### 3.3. Thermal Stability and Thermal Conductivity

The thermal stability of Al_2_O_3_/PU and PDA-Al_2_O_3_/PU composites is analyzed and shown in Figure 8. Micro-quotient thermogravimetric analysis (DTG) refers to measuring the relationship between the weight loss rate of a sample and temperature.

Obviously, both pure PU and composites exhibit two weight loss temperature ranges; 210–320 °C is the degradation of polyurethane in the hard section and 360–430 °C is the degradation of polyol in the soft section. It can be seen that with the increase of particles, the initial decomposition temperature of the composite material increases, and the initial decomposition temperature of PDA-Al_2_O_3_/PU is higher than that of Al_2_O_3_/PU. Moreover, the weight loss rate of the PDA-Al_2_O_3_/PU composites increases during the first and second weight loss stages, and the decomposition temperature corresponding to the maximum decomposition rate is higher than that of pure PU and Al_2_O_3_/PU composites. With the increase of the filler content, the weight loss rate of the composites in the first and second weight loss stages increases, and the decomposition temperature corresponding to the maximum decomposition rate increases. This indicates that the addition of particles improves the thermal stability of the composites. This is because Al_2_O_3_ modified by PDA can be uniformly dispersed into the PU matrix, which enhances the interaction between PDA-Al_2_O_3_ and PU, and the increase of PDA-Al_2_O_3_ makes the overall structure of the composites more compact. Then, due to its excellent high thermal conductivity, Al_2_O_3_ particles can reduce the local accumulation of heat, reduce the thermal stress in the PU matrix, and slow down the damage of temperature to the PU matrix. Further, the thermal stability of 30 wt% PDA-Al_2_O_3_/PU composite reaches the highest level.

Figure 9 shows the thermal conductivity of Al_2_O_3_/PU and PDA-Al_2_O_3_/PU composites.

The results show that the thermal conductivity of the composites increases with the fraction of particles. Compared with Al_2_O_3_/PU composites, PDA-Al_2_O_3_/PU composites have higher thermal conductivity under the same particle content. This can be attributed to the polydopamine-modified Al_2_O_3_ surface grafted with more functional groups, which enhanced the interaction and compatibility with PU and further improved the dispersion in the PU matrix. At low loadings, the particles are surrounded by the PU matrix and cannot contact each other. As the amount of particles increases, more and more heat conduction paths are formed in the PU matrix, so the thermal conductivity of the composites is constantly improved. The thermal conductivity of 30 wt% PDA-Al_2_O_3_/PU composite reaches the highest level, which is 138% higher than that of pure PU.

### 3.4. Dynamic Mechanical Analysis

The storage modulus (E’) reflects the energy stored during the deformation of the material due to elastic deformation, that is, the rigidity of the material. The tanδ curve is obtained from the ratio of the loss modulus to the storage modulus. The highest peak of this curve is defined as glass transition temperature (Tg). The glass transition temperature (Tg) is the temperature at which the resin transitions from a glassy state to a highly elastic state. Above this temperature, the polymer exhibits elasticity. Therefore, the operating temperature of polyurethane is above the glass transition temperature. Figure 10 shows the relationship between storage modulus (a) and loss factor (b) of Al_2_O_3_/PU and PDA-Al_2_O_3_/PU composites with respect to temperature.

With the increase of the particles, the storage modulus of the composites continues to increase, and the addition of PDA-Al_2_O_3_ can better improve the storage modulus of the composites than Al_2_O_3_ in Figure 10a. In addition, with the increase of temperature, the storage modulus in the glass rubber transition zone suddenly decreases, which is mainly due to the energy dissipation phenomenon caused by the coordinated movement of the polymer chain. Firstly, high-modulus Al_2_O_3_ can effectively improve the storage modulus of composites [7]. Secondly, the addition of PDA-Al_2_O_3_ will enhance the bonding strength of the matrix interface, inhibit the movement of the polymer chain, and effectively transfer the stress between PU and PDA-Al_2_O_3_, thereby improving the stiffness of composites. Figure 10b shows that compared to pure PU, the peak height of tanδ decreases after the addition of particles. The peak height of PDA-Al_2_O_3_/PU is lower than the peak height of Al_2_O_3_/PU, and Tg gradually transfers to a higher temperature. The decrease in tanδ peak height and the right shift of Tg mean less chain mobility, and the interface strength is stronger [39]. This further proves the positive effect of particle modification on the interfacial force between the PU matrix and particles. Table 1 shows the glass transition temperatures of composites.

### 3.5. Tensile Properties

Figure 11 shows the tensile properties of Al_2_O_3_/PU and PDA-Al_2_O_3_/PU composites.

Within the material’s elastic limit, the ratio of stress to strain is called the material’s Young’s modulus, which is a physical quantity that characterizes the material’s resistance to elastic deformation and reflects the rigidity of the material [16]. Similar to the storage modulus, the stress–strain ratio shows an increase in Young’s modulus, which indicates that the composite is more rigid than pure PU.

Figure 11a–c shows that the 30 wt% PDA-Al_2_O_3_/PU has a significantly higher Young’s modulus, tensile strength, and elongation at break than pure PU. Compared with pure PU, Young’s modulus is increased by 95%, tensile strength is increased by 128%, and the elongation at break is increased by 76%. This is due to the high modulus and high hardness of Al_2_O_3_, which makes PU have stronger resistance to deformation and high hardness. The addition of particles strengthens and toughens the pure PU. However, as the content of particles increases, the mechanical properties of the composites increase slightly, and when the content of particles is low, the mechanical properties of Al_2_O_3_/PU and PDA-Al_2_O_3_/PU are almost equivalent. However, when the content of particles is 30 wt%, the mechanical properties of PDA-Al_2_O_3_/PU are slightly higher than those of Al_2_O_3_/PU. This is due to the presence of a large number of phenolic hydroxyl groups on the surface of PDA-Al_2_O_3_, it has a strong interaction with isocyanate groups and urethane groups in the PU matrix. This is consistent with the SEM analysis results. The results show that it is suitable for high-strength and toughness composites. In short, the 30 wt% PDA-Al_2_O_3_/PU composite has the highest mechanical properties.

### 3.6. Water Absorption

A humid environment will also affect the performance of electronic packaging to some extent [36]. When electronic packaging absorbs too much moisture, the insulation property of the composites decreases, which affects the service life. Figure 12 shows the water adsorption of Al_2_O_3_/PU and PDA-Al_2_O_3_/PU composites.

Due to the free volume between the polymer molecules and the presence of polar groups, water molecules can diffuse and penetrate faster in the polymer. When the number of particles increases, the accumulation of molecular chains becomes tighter and the degree of intermolecular cross-linking increases, resulting in an increase in the permeation resistance; therefore, the water absorption rate decreases. Compared with Al_2_O_3_/PU, the active surface of PDA-Al_2_O_3_ is combined with polar functional groups in PU, which results in the reduction of polar functional groups, thereby reducing the water absorption to a certain extent. However, as the particle content increases to a certain value, the degree of intermolecular cross-linking no longer increases significantly, so under a 30 wt% load, the water absorption of Al_2_O_3_/PU composites and PDA-Al_2_O_3_/PU composites becomes more similar.

## 4. Conclusions

Polyurethane composites with alumina were prepared, and their mechanical and thermal properties were compared. Compared with pure PU, the thermal conductivity of a 30 wt% PDA-Al_2_O_3_/PU composite was improved by 138%. The glass transition temperature curve of the composites prepared from it with alumina were shifted to a higher temperature than that of pure PU, and the storage modulus was also improved. The dispersion of the particles in pure PU was investigated using SEM. When it was filled with polydopamine-modified alumina, the results showed better dispersion compared to pristine alumina. The thermal stability of the composites was higher than that of pure PU. Compared with pure PU, the 30 wt% PDA-Al_2_O_3_/PU had 95% more Young’s modulus, 128% more tensile strength, and 76% more elongation at break than pure PU. The 30 wt% PDA-Al_2_O_3_/PU had the lowest water absorption. With the addition of alumina, the comprehensive properties of composites have been greatly improved.

## Figures and Tables

**Figure 1 materials-13-01772-f001:**
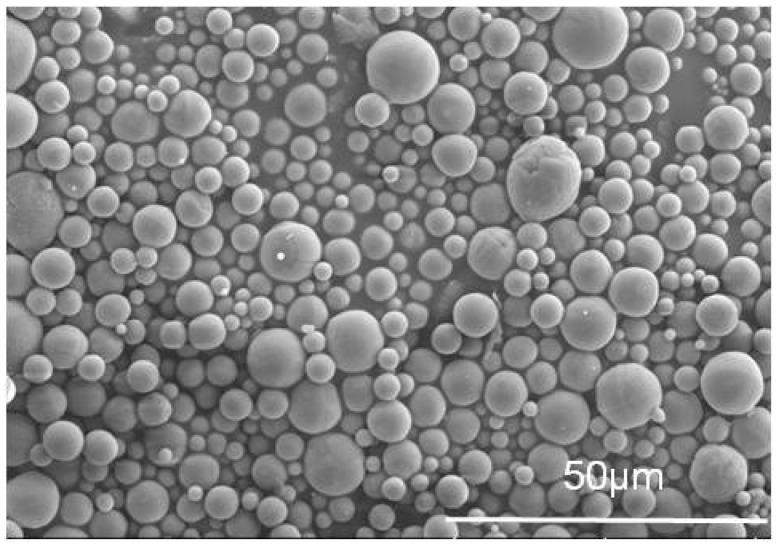
SEM images of pristine Al_2_O_3_.

**Figure 2 materials-13-01772-f002:**
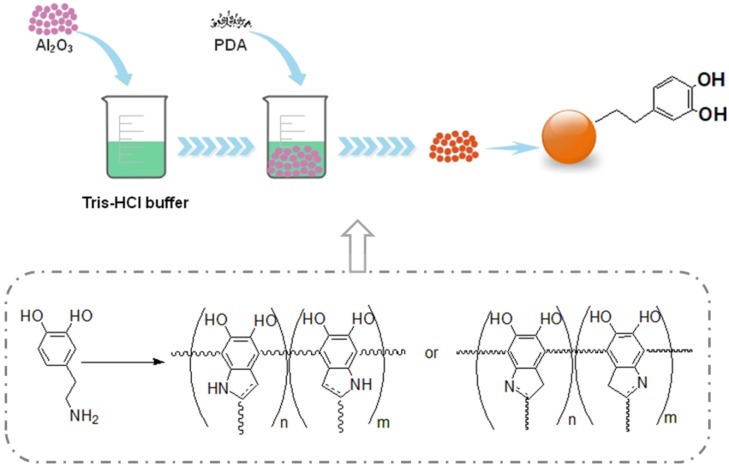
Scheme of modification of pristine Al_2_O_3_ surface using polydopamine (PDA).

**Figure 3 materials-13-01772-f003:**
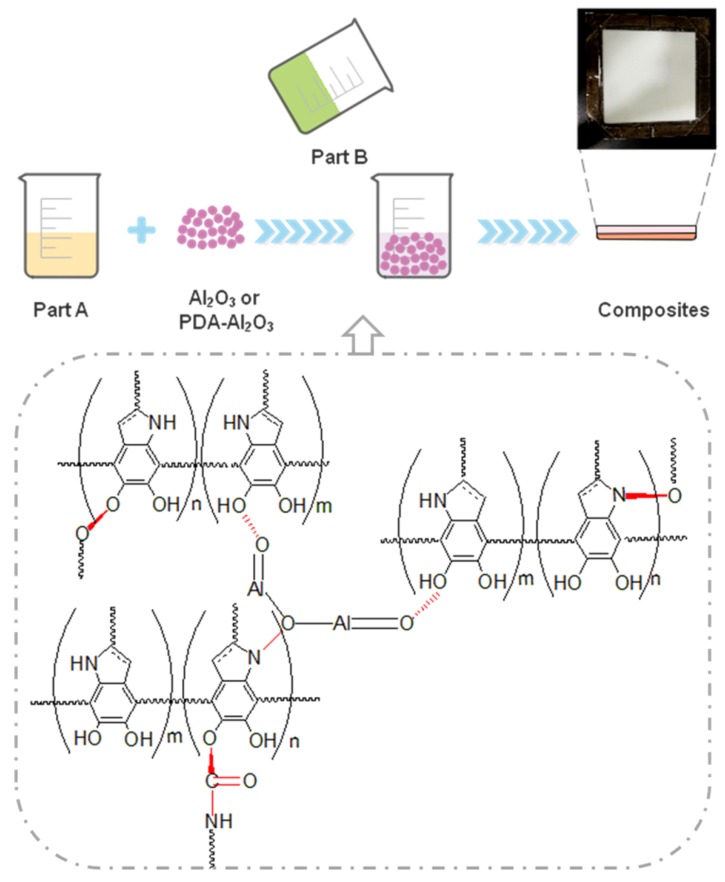
Preparation of polyurethane (PU) composites filled with Al_2_O_3_ and PDA-Al_2_O_3_.

**Figure 4 materials-13-01772-f004:**
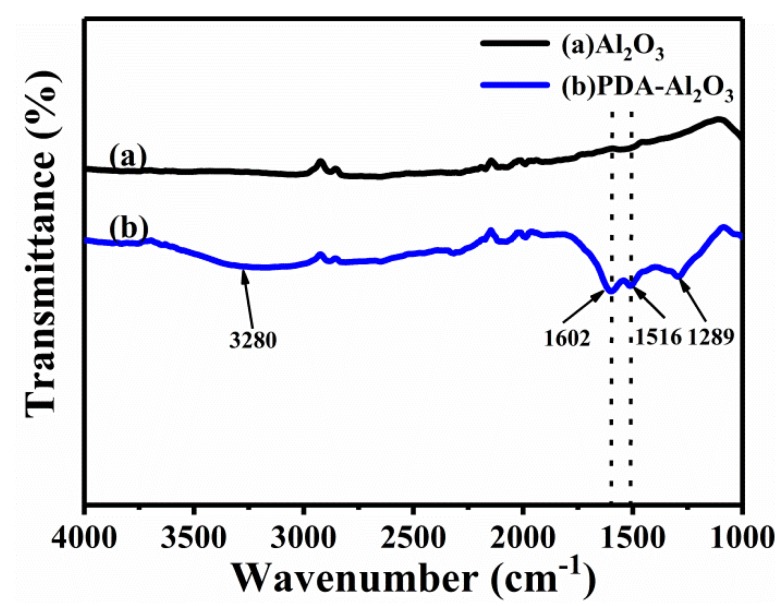
FTIR results for Al_2_O_3_ and PDA-Al_2_O_3_.

**Figure 5 materials-13-01772-f005:**
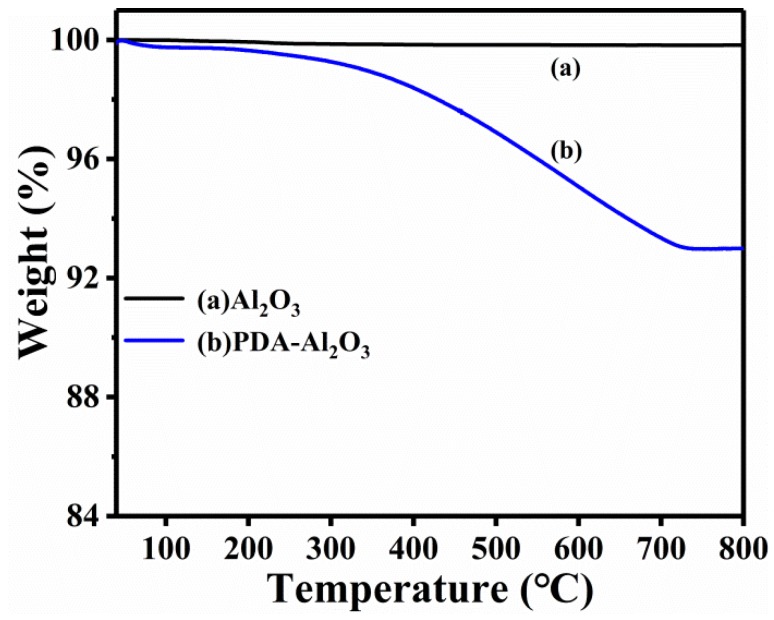
Thermogravimetric analysis (TGA) results for Al_2_O_3_ and PDA-Al_2_O_3_.

**Figure 6 materials-13-01772-f006:**
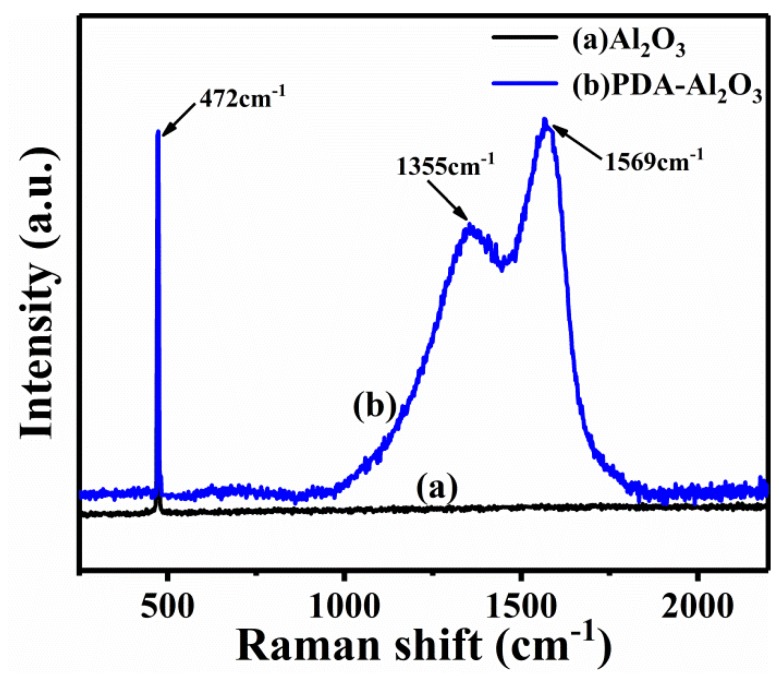
Raman results for Al_2_O_3_ and PDA-Al_2_O_3_.

**Figure 7 materials-13-01772-f007:**
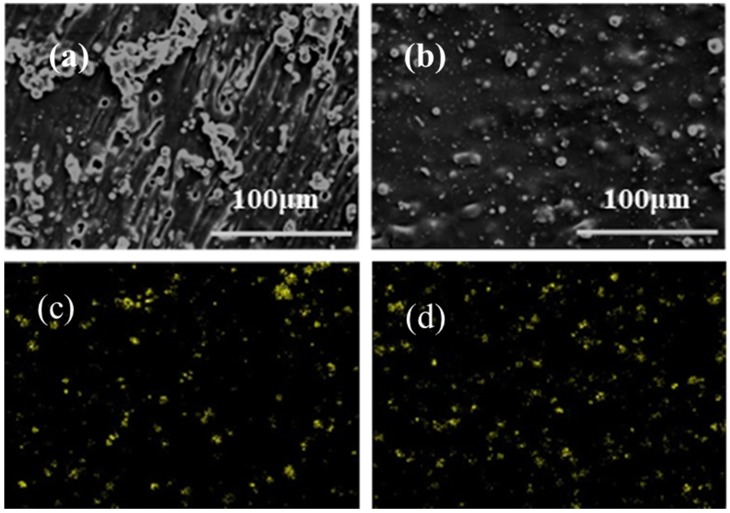
SEM morphology of composites filled with (**a**) 30 wt% Al_2_O_3_, (**b**) 30 wt% PDA-Al_2_O_3_, EDS element distribution of yellow spots in (**c**) 30 wt% Al_2_O_3_/PU, and (**d**) 30 wt% PDA-Al_2_O_3_/PU.

**Figure 8 materials-13-01772-f008:**
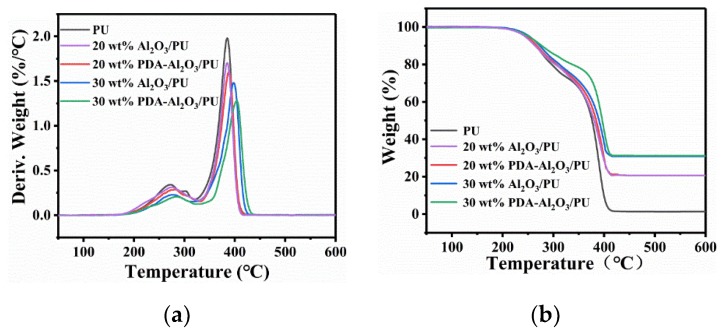
Thermal stability of Al_2_O_3_/PU and PDA-Al_2_O_3_/PU composites: (**a**) TG, (**b**) DTG curves.

**Figure 9 materials-13-01772-f009:**
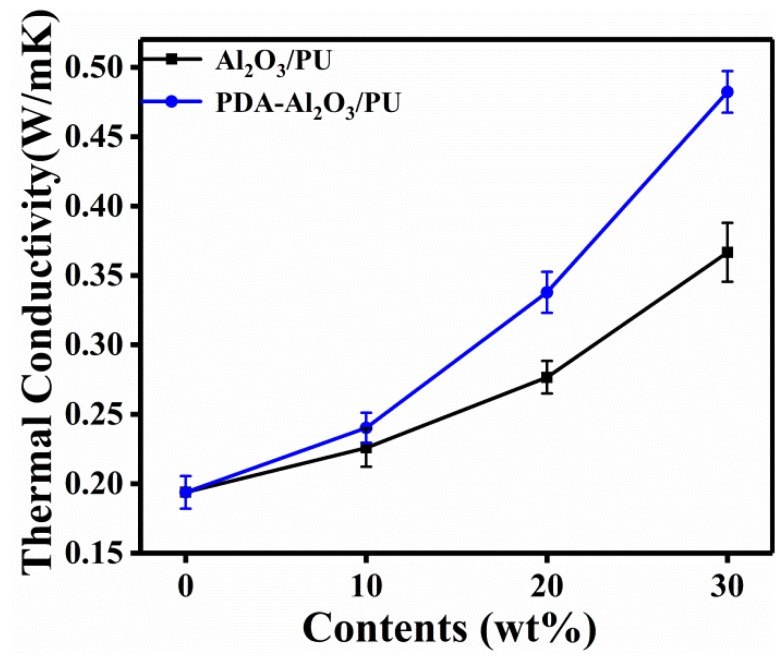
Thermal conductivity of Al_2_O_3_/PU and PDA-Al_2_O_3_/PU composites.

**Figure 10 materials-13-01772-f010:**
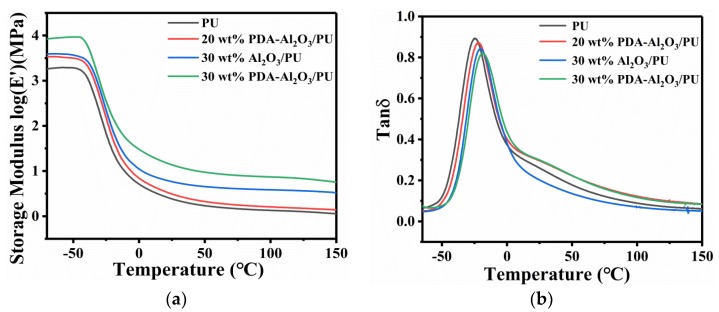
The relationship between storage modulus (**a**) and loss factor (**b**) of Al_2_O_3_/PU and PDA-Al_2_O_3_/PU composites with respect to temperature.

**Figure 11 materials-13-01772-f011:**
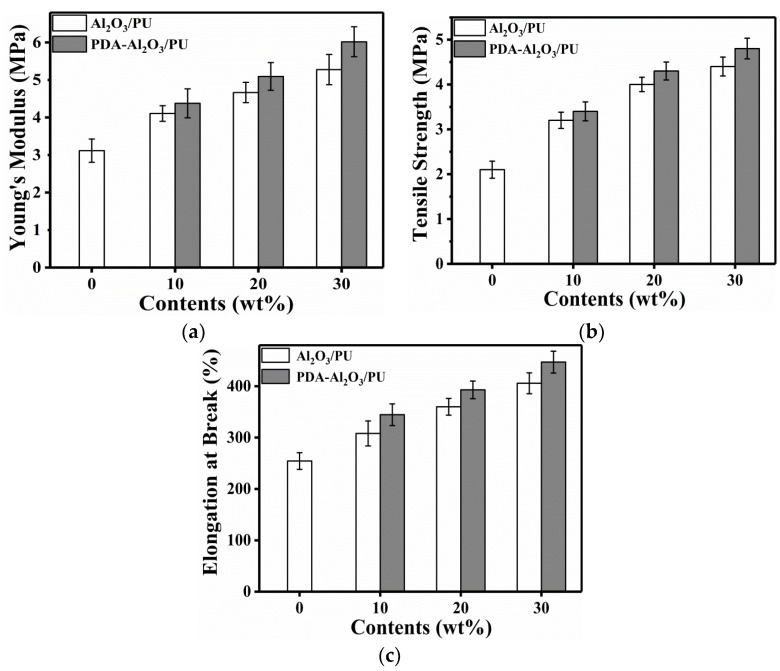
Mechanical properties of the Al_2_O_3_/PU and PDA-Al_2_O_3_/PU composites: (**a**) Young’s modulus, (**b**) tensile strength, and (**c**) elongation at the break.

**Figure 12 materials-13-01772-f012:**
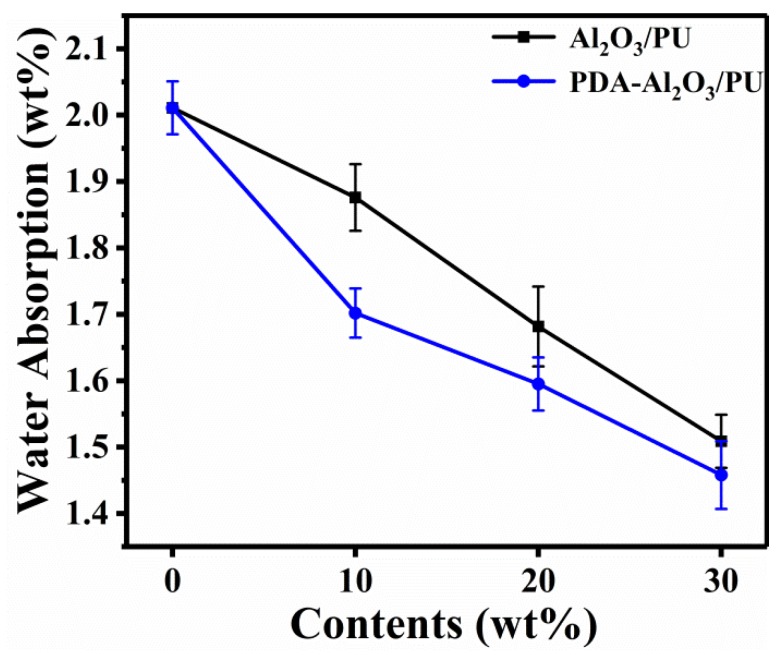
Water absorption of Al_2_O_3_/PU and PDA-Al_2_O_3_/PU composites.

**Table 1 materials-13-01772-t001:** Glass transition temperature of the composites.

Content	PU	20 wt% PDA-Al_2_O_3_/PU	30 wt% Al_2_O_3_/PU	30 wt% PDA-Al_2_O_3_/PU
Tg (°C)	−24.7	−22.3	−20.1	−18.6

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
