# Peer review of "Polydopamine-Modified Al2O3/Polyurethane Composites with Largely Improved Thermal and Mechanical Properties"

_materials, 2020, doi:10.3390/ma13071772_

Round 1

Reviewer 1 Report

Manuscript ID: materials-734091   The study provides a new approach to enhance thermal conductivity of polyurethate based substrates by using Al2O3 particles coated with Polydopamine (PDA) which easily adheres to inorganic and organic materials surface, for the preparation of Alumina/polyurethane composites. The cost of polydopamine and the other auxilliary substances should be mentioned in relation to such a broad application.   As a general remark, the work is interesting but scientifically needs further support and from the langauge aspect it needs editorial assistance.   Specific comments   Line 12 Explain abbreviation 'TIMS'. L. 13. Replace 'evaluates' by 'evaluated' L. 16. % goes before w/w. Correct in other parts of the m/s  L. 38. Replace 'such as' with 'are' L. 39. Change 'grapheme' to 'graphene' L. 57. Replace 'such as' with 'are' L. 72. Delete 'have' L. 90. Explain 'tris' L. 99. 2.3 Preparation ...' The whole paragraph need re-writing.  L. 101. what is part A? polyurethane? it is not clear L. 105. Please provide consistency of PDA-Al2O3/PU. What is %PDA, AL2Oe, PU L.126. Please give calculatio formula for the mechanical properties L. 140. Change 'Results' to 'Results and Discussion'  L. 145. Delete 'significantly' L. 151. Delete 'characterised'  L. 154, 155. Move text 'Due to its ... 800 oC' to the ende of the sentence after 7.01%.  L. 168. Change 'measured; to 'taken'. The particles in Fig 6 were spherical. What was the shape of the other composite particles? The effect of particle shape on the mechanical properties is very important (please quote ref. Nikolakakis and Pilpel, Powder Technology,42 (1985) 279 - 233). L. 173. Remove 'seriously' L. 177 Replace 'of dopamine' with 'with polydopamine' L. 179 Give PDA-Al2O3/PU  compositin clearly in the text. L. 183. Does it mean as particle number increase? L. 185. All curves drop at about 200-250 oC. How do you explain the shoulder of green line (what is 30% in the composition PDA-Al2O3/PU?). L. 206. These two variables (storage modulus  and loss factor) have to be explained better: what is their physical meaning? how are they computed?   L. 213. Corect Figure 10a L. 252. How to you control the effect of particle shape? At such small particle sized is of paramount importance (Nikolakakis and Pilpel 1985 Powder Technology 42, 279 - 233). L. 267. Add 'number' before particles L. 273. Did you try higher water content levels? L. 274. Replace 'Discussion' with 'Conclusions'           

Reviewer 2 Report

Reviewer Comments for Materials 734091 “Composites with Largely Improved the Thermal and Mechanical Properties”

The authors present a research study where they compare the materials properties of a polyurethane filled with Al2O3 and Al2O3 modified with polydopamine. The purpose of this study is to improve the materials properties of insulating materials for the packaging industry. The modification of the Al2O3 filler, the polymer-filler interaction could be improved over unmodified fillers. The methodology to measure materials properties was sufficient, assessment and interpretation of data could be improved.

I have some suggestions that the authors may want to address in order to improve the quality and readability of this manuscript.

  1. Abstract: please explain TIMs
  2. Abstract: DMA means dynamic mechanical analysis, not dynamic thermomechanical performance. Please correct.
  3. Abstract: glass transition shifts to the right. I would rather write that it shifts to higher temperatures.
  4. Line 44: I guess you are talking about electrical insulating? Please clarify as this is not fully clear. Insulating could mean thermally insulating too.
  5. Lines 78-79: DMA see point 2
  6. Section 2.2 please give more details (e.g. what is the water volume used?)
  7. Section 2.4. please give more details, e.g. FTIR: was the measurement performed in transmission as ATR? Resolution? The number of scans? Tensile tests: was the dumbbell shape also from a standard? What was the sample size for DMA?
  8. For all measurements (except water absorption): How many samples were measured? What statistical methods were used? Descriptive statistics (mean/median and standard deviation)? Please add. In some graphics, error bars are shown, but no comment on the sample size or statistics are given. For all other measurements, figure caption should mention if a representative measurement is displayed or if this was just an N=1 measurement. In that case, no conclusions can be made.
  9. Section 3: I would rather call it “Results and Discussion” and section 4 “Conclusion”.
  10. Figure 3: A comparison to PDA alone might be interesting too. Also, you may want to consider a zoom into the area of interest (as an inset or so).
  11. Figures 4 and 8: TGA curves would benefit from the derivative of the mass loss. This would make it easier to see the onset of mass loss and distinguish between different regions of decomposition.
  12. Line 154: Do the authors mean Al2O3 instead of PDA-Al2O3?
  13. Figure 6 is not referenced in the text. What is its purpose?
  14. Figure 7: In the caption, you mention that a and c show unmodified, and b and d modified filler. In the main text, you say differently. Please check and correct.
  15. Figure 8 interpretation is not very clear and the statements you make can not be verified from the figure. See #11, derivative would help here. Why do you compare PU, 20 wt% PDA-Al2O3, 30 wt% Al2O3 and PDA-Al2O3? Could you please give the rationale for this? Showing 20 wt% Al2O3, in addition, would also make sense here.
  16. Figure 10: Storage modulus is usually displayed in logarithmic scale. Please revise. Also, the explanation and conclusion you make from the measurements are not clear. Only the storage modulus in the glassy state can be evaluated with your graphic. And since Tg is below freezing, and thus below working temperature of the material as electrical insulator, this says nothing about its properties. Also, the damping at this temperature is not impacting the use of the material unless it is mainly used around -25 C. Also, the shift in Tg is marginal (please add numbers) and has not really an influence on the damping. The same is true for the decrease in peak height. The damping also takes the whole area under the curve into account, not only the peak max. An increase in Tg means less chain mobility, which is most likely due to improved filler-matrix interaction. Please revise the whole paragraph.
  17. Figure 11: the 2 bars for 0 wt% make no sense as it is the same material when there is no filler added. So just one bar should be displayed here. For all individual filler contents, the error bars are overlapping. Thus, there is technically no difference. If anything, you can talk about a trend, but this is nothing significant. Unless you have performed a T-test or similar and can justify that there is a significant difference. But since you have not reported any statistics at all, I assume that this is not the case. Please rephrase and be careful with your conclusions.
  18. Line 252: Please add a reference for the statement that Al2O3 has a high hardness and modulus.
  19. Figure 12: Same as for Figure 11: there are no differences for the 20 and 30 wt% filler contents because the error bars are overlapping. Thus, it is not correct to state that 30 wt% of PDA-Al2O3 has the lowest water absorption. Please rephrase.

Round 2

Reviewer 1 Report

Authors have made a good effort to improve the presentation of their work. However, since secret formulas do not usually allowed to pass by by most journals, I don't propose acceptance of the paper unless basic chemical information of the polyurethane, part A together with the commercial name, manufacturer are provided. unless the final decision rests with the Editorial. 

Additionally, comments and explanation for the lack of particle shape examination should be added as part of the experimental design.  

Reviewer 2 Report

see attachment
